# Distribution and Polarization of Caries in Adolescent Populations

**DOI:** 10.3390/ijerph18094878

**Published:** 2021-05-03

**Authors:** Helen Schill, Uta Christine Wölfle, Reinhard Hickel, Norbert Krämer, Marie Standl, Joachim Heinrich, Jan Kühnisch

**Affiliations:** 1Department of Conservative Dentistry and Periodontology, School of Dentistry, Ludwig-Maximilians-Universität München, 80336 Munich, Germany; helenschill@gmail.com (H.S.); uta.woelfle@med.uni-muenchen.de (U.C.W.); hickel@dent.med.uni-muenchen.de (R.H.); 2Department of Paediatric Dentistry, Medical Centre for Dentistry, University Medical Center Giessen and Marburg, Campus Giessen, 35392 Giessen, Germany; Norbert.Kraemer@dentist.med.uni-giessen.de; 3Institute of Epidemiology, Helmholtz Zentrum München—German Research Centre for Environmental Health, 85764 Neuherberg, Germany; marie.standl@helmholtz-muenchen.de (M.S.); Joachim.Heinrich@med.uni-muenchen.de (J.H.); 4Institute and Outpatient Clinic for Occupational, Social and Environmental Medicine, University Hospital of Munich, Ludwig-Maximilians-Universität München, 80336 Munich, Germany; 5Allergy and Lung Health Unit, Melbourne School of Population and Global Health, The University of Melbourne, Melbourne, Victoria 3010, Australia

**Keywords:** epidemiology, caries, distribution pattern, prevalence, caries polarization

## Abstract

The aim of this study was to determine the proportion of adolescents with severe caries to analyze the prevalence of caries and to visualize the unequal distribution. Data from three epidemiological studies (10- and 15-year-olds: GINIplus and LISA cohorts in Munich, Bavaria; 12-year-olds: LAGZ survey in Bavaria, Germany) with 2875 adolescents were available for analysis. All individuals were examined according to the WHO standard. Statistics included the calculation of mean dmft/DMFT values (standard deviation), Significant Caries Index (SiC) values, Specific Affected Caries Index (SaC) values, and Lorenz curves. Overall caries-free status was 58.6% in primary and 83.9% in secondary teeth (10-year-olds), 61.5% (12-year-olds), and 64.6% (15-year-olds). The proportion of 12- and 15-year-olds with at least four DMFTs was 9.4% and 8.3%, respectively. In addition, eight 15-year-olds with DMFT values ≥8 (0.6%) were registered. The SaC/SiC values amounted to 1.8/0.9 DMFT (10-year-olds), 2.6/2.8 DMFT (12-year-olds), and 2.5/2.5 DMFT (15-year-olds). The mean DMFT values in the upper 1% of subjects were 4.2 DMFT (10-year-olds), 8.5 DMFT (12-year-olds), and 8.5 DMFT (15-year-olds). Thus, caries is not equally distributed throughout adolescence, but individuals with severe caries are rare. Nevertheless, further interdisciplinary research seems to be needed to clarify potential risk factors.

## 1. Introduction

In recent decades, many industrialized nations have seen a remarkable decline in prevalence, experience, and incidence of caries [1,2,3]. However, despite this encouraging trend, caries remains a global burden, and approximately 3.5 billion individuals remain with both untreated caries and the associated challenges [4,5]. In Germany, the documented prevalence and experience of caries has decreased constantly over time, reaching an all-time low, especially in younger generations that have probably benefitted substantially from individual- and group-based preventive programs. In detail, the proportion of caries-free subjects reached 81%, and the average decayed, missing, and filled permanent teeth (DMFT) value amounted to 0.5 in 12-year-olds in 2014 [6]. Nevertheless, although the population-wide burden of caries has fallen, it is also a well-known epidemiological trend that caries is not equally distributed over the population, typically affecting groups or individuals at elevated risks [7,8,9,10].

Contrary to the substantial improvement in dental health, we diagnosed several individuals during the past decade in clinical dental practice whose permanent dentition was in exceptionally poor condition (Figure 1) and required extensive dental care. A common clinical characteristic in all cases was the generalized appearance of active caries lesions, ranging from demineralization to gross cavitation in a substantial number of permanent teeth, which can be compared to a certain extent with the well-known phenomenon of early childhood caries in primary dentition [11]. Recognizing that consultations regarding adolescents with severely affected permanent dentition might currently be exceptional in German dental practices [6,9], it appears to be of relevance to determine the proportion and extent of the hypothesized unequal distribution from an epidemiological perspective to ensure these patients are also tangible in epidemiological studies. The aim of this study was therefore to investigate the proportion of adolescents with severe caries, to analyze the prevalence of caries, and to highlight the uneven distribution of caries. The methodological modernization undertaken in this study combined the use of existing statistical measures, e.g., the SiC index combined with narrower thresholds, such as 20%, 10%, 5%, 2%, and 1%, the SaC index, and Lorenz curves, to illustrate the uneven distribution of caries in the adolescent population.

## 2. Materials and Methods

### 2.1. Study Populations and Ethical Approval

The data set used originates from three previously conducted epidemiological studies, which included essentially healthy individuals: (1) 10-year follow-up of the prospective population-based Munich birth cohort study, including healthy, mature newborns of German origin with a birth weight above 2500 g (GINIplus and LISA, 2005–2009); (2) 15-year follow-up of the same Munich birth cohort study (GINIplus and LISA, 2010–2014); and (3) cross-sectional survey study on dental health throughout Bavaria including 12-year-old school children (6th grade) in 107 secondary schools (LAGZ, 2016). Details regarding each study background, inclusion and exclusion criteria, and the recruitment strategy of the GINIplus and LISA birth cohorts [12,13] and the LAGZ survey [12] were published previously. For each epidemiological study, ethical approval was received from the corresponding ethical boards (10-year-olds: Bavarian Board of Physicians No. 05100, No. 07098; 12-year-olds: Ethics Commission at the University of Gießen AZ 94/15 and Bavarian Ministry of Education and Religious Affairs X.7-BO4106/482/71; 15-year-olds: Bavarian Board of Physicians No. 10090 and No. 12067). Written authorization was obtained from all participating adolescents and their legal guardians. All dental examinations were performed in accordance with the ethical standards of the Institutional Research Board and the modified Helsinki declaration [14]. Reporting propositions of the STROBE guidelines for observational studies were applied [15].

### 2.2. Dental Examinations

After brushing their teeth, if necessary, all subjects were investigated by each of the responsible examiners using standard instruments: (1) a dental mirror; (2) a blunt Community Periodontal Index (CPI) probe (CP-11.5B6, Hu-Friedy, Chicago, IL, USA); (3) a mobile examination lamp (GINIplus/LISA: Ri-Magic, Rudolf Riester GmbH, Jungingen, Germany; LAGZ: Haeberle Halux 50S, 50 Watt, Haeberle GmbH, Stuttgart, Germany); and (4) cotton rolls (GINIplus/LISA) or compressed air (LAGZ). X-ray examinations were not carried out. All primary (10-year-olds only) and permanent teeth (10-, 12- and 15-year-olds) were evaluated for their caries and restoration status, according to the principles of the dmft/DMFT index [16]. A cavitated caries lesion was recorded when the tooth surface had an unmistakable cavity, undermined enamel, or a detectably softened floor or wall [16]. Non-cavitated carious lesions were not considered in the present analysis. Restoration was documented only when applied for caries-related reasons. When restorations were located for other reasons, e.g., molar-incisor hypomineralization, they were not recorded as part of the dmft/DMFT index. Other enamel defects, e.g., hypoplasia, fluorosis (diffuse opacities), amelogenesis/dentinogenesis imperfecta, erosion, tooth wear, trauma-related dental defects, restorations, and sealants, were not registered.

### 2.3. Calibration

The calibration of the study teams, consisting of examiners and principal investigators, has been extensively described previously [13,17,18]. Prior to all studies, theoretical information about the study design, diagnostic principles, and indices was provided. In addition, training consisted of analysis and discussion of high-resolution photos of single tooth surfaces with various clinical findings, e.g., (non-)cavitated carious lesions, restorations, and fissure sealants, in addition to potential differential diagnoses, e.g., molar-incisor hypomineralization, erosion, or fluorosis. After the methodical training session, the routines of dental examinations were exercised clinically in several patients by all involved examiners under supervision of the principal investigators. Intra- and interexaminer reproducibility was measured for all examiners and found to be sufficient [12,17,19,20].

### 2.4. Statistical Analysis

Data collection and analyses were carried out using a database system (Access 2010, Microsoft Corporation, Redmond, WA, USA) and Excel spreadsheets (Excel 2010, Microsoft Corporation, Redmond, WA, USA). Mean values and standard deviations of the dmft/DMFT index and its components were computed to describe the caries experience. Explorative statistical analyses included Lorenz curves [21], the Significant Caries Index (SiC), and the Specific Affected Caries Index (SaC). The SiC was calculated on the basis of the mean DMFT values and reflects the upper third of the most caries-affected subjects [22]. In addition to the upper-third threshold, we calculated the mean values and standard deviations of the dmft/DMFT index and its components for the 20%, 10%, 5%, 2%, and 1% thresholds. Additionally, the SaC was calculated [23]. The SaC represents the mean dmft/DMFT value for the proportion of subjects with at least one decayed, filled, or extracted tooth (dmft/DMFT > 0).

## 3. Results

The present analysis included dental records from 2875 patients. The study population characteristics are summarized in Table 1. Data about the prevalence, experience, and polarization of caries are presented in Table 2 and Table 3. In detail, the proportion of caries-free adolescents (dmft/DMFT = 0) among 10-year-olds was 58.6% in primary dentition and 83.9% in permanent dentition. The proportion of caries-free permanent dentition (DMFT = 0) was 61.5% in 12-year-olds and 64.6% in 15-year-olds. The mean experience of caries was 0.3 DMFT, 1.0 DMFT, and 1.0 DMFT in 10-, 12- and 15-year-olds, respectively. When analyzing the components of the DMFT index, it was consistently shown that the FT component was highest and amounted to ~100% in 10-year-olds, ~80% in 12-year-olds, and ~90% in 10- and 15-year-olds.

To identify the proportion of individuals with the highest burden of caries, cut-off values (dmft/DMFT ≥ 4 and dmft/DMFT ≥ 8) were used (Table 2). The proportion of 12- and 15-year-olds with at least four DMFTs was 9.4% and 8.3%, respectively. In addition, eight 15-year-olds with DMFT ≥ 8 (0.6%) were registered (Table 2), which can be linked to the presence of an extreme caries burden. Caries-affected children without restorations in primary dentition (dmft/DMFT > 0 and ft/FT = 0) comprised up to 6.5% (10-year-olds); the corresponding proportions for the permanent dentition were 1.1% (10-year-olds), 5.0% (12-year-olds), and 3.1% (15-year-olds). Six individuals had a DT value above 4 without having any other restorations (0.05%). In one 15-year-old patient, seven carious teeth were detected without the presence of any restoration.

The SaC and SiC illustrate a more detailed view of caries polarization (Table 2). The SaC was 1.8 DMFT (10-year-olds), 2.6 DMFT (12-year-olds), and 2.5 DMFT (15-year-olds). With a similar aim, the SiC determined the average experience of caries as the mean DMFT of the upper third most affected by caries [22]. These values were 0.9 DMFT (10-year-olds), 2.8 DMFT (12-year-olds), and 2.5 DMFT (15-year-olds). Additionally, the experience of caries for the upper 20%, 10%, 5%, 2%, and 1% is shown in Table 3. The mean DMFT values in the upper 1% group of subjects was 4.2 DMFT (10-year-olds), 8.5 DMFT (12-year-olds), and 8.5 DMFT (15-year-olds). The Lorenz curves (Figure 2) illustrate an unequal distribution of the experience of caries and the components of the dmft/DMFT index across the three populations. When considering a cumulative patient proportion of 80%, the cumulative burden of caries corresponded to 25% in 12-year-olds and 19% in 15-year-olds. Accordingly, a minority of subjects accounted for the majority of the caries burden (Figure 2).

## 4. Discussion

On the basis of WHO global goals for oral health in 2020 [24], Germany established firm ambitions to reach a DMFT value below 1 among 12-year-olds [25]. This goal was confirmed with a DMFT value of 0.5 and 81% caries-free 12-year-olds in a previously conducted, population-based, and representative German Oral Health study [6]. The comparable oral health parameters for 12-year-olds from the present data analysis were found to be lower (DMFT value of 1.0; 61.5% caries-free adolescents, Table 2). The documented values for 10-year-olds (DMFT 0.3; 83.9% caries-free adolescents) and 15-year-olds (DMFT value of 1.0; 64.6% caries-free adolescents) can be ranked in the same order of magnitude and are mostly comparable to results from other German epidemiological trials [12,25,26,27]. Essentially, the epidemiological data indicate a low experience of caries for the investigated populations. Furthermore, throughout all three populations, the FT component of the dmft/DMFT index was consistently high in comparison to the DT component (Table 2). The MT component was found to be negligible.

Considering that the aim of the present study was to determine the proportion of individuals with a high caries burden, several aspects need to be discussed. In general, on the basis of the present data (Table 2 and Table 3, and Figure 2), caries was unequally distributed in all investigated populations. In detail, a substantial proportion of adolescents were classified as caries-free in terms of the dmft/DMFT index, and a minority of subjects accounted for the majority of the caries burden (Figure 2). The same tendency of unequal distribution needs to be highlighted for adolescents with any experience of caries. The majority of these subjects—approximately three-quarters of 12- and 15-year-olds with a DMFT > 0—had a DMFT from 1 to 3 (Table 2). When considering the experience of caries for adolescents with DMFT > 0 (SaC index) or for the upper third of subjects with the highest caries burden (SiC index), both groups showed substantially higher DMFT values in comparison to the documented mean values (Table 2). When analyzing data for individuals with the 1% threshold of the SiC index, it became evident that prevalence of caries in this small subgroup was eight times higher (Table 2). The number of subjects with high (DMFT ≥ 4) and extreme experiences of caries (DMFT ≥ 8) decreased with increasing DMFT values. In addition, the proportion increased with growing age. Overall, nine individuals were detected with severely decayed permanent dentition with a DMFT ≥ 8 (10-year-olds: N = 0; 12-year-olds: N = 1; 15-year-olds: N = 8). However, regarding the very small number (N = 9) of severely caries-affected adolescents compared to the overall investigated population (N = 2875), these individuals represent adolescents with the poorest oral health status.

Contrary to the potential epidemiological negligibility of adolescents with severely destructed dentitions in the investigated populations, the following significant clinical consequences for each individual need to be considered: loss of the anatomical form and functionality of several permanent teeth more or less immediately after tooth eruption, painful pulpal or periodontal complications, and time-consuming and expensive restorative or surgical treatment needs (Figure 1). Caries etiology is usually a complex and multifactorial field. However, in dental practice, we empirically recorded a mainly carbohydrate-rich diet with frequent intake of cariogenic/erosive beverages, and/or processed nutrients and neglected oral hygiene, which basically corresponds to the well-known caries etiology of ecological plaque hypothesis [28,29]. Individual factors, such as psychological and behavioral factors of parents and socioeconomic status [30], which are known to play a role in the experience of caries, were not continuously recorded in the studies. Consequently, no clear statement can be made regarding their influence. Regardless of the individual risk factors, it has been previously documented that both dental anxiety and oral health-related quality of life are co-dependent factors affecting a person’s general well-being [31,32,33]. In addition, we suspect dental neglect, conspicuous behavior, or possible psycho-emotional disturbances of any origin are possible co-variables in some cases. This probably negatively influenced dental awareness and adherence to recommended preventive or operative treatment procedures.

Consequently, patients discontinued initiated treatments, which may indicate that any dental treatments were generally rejected and appointments appeared to be challenging for both patients and dental professionals. Therefore, interdisciplinary management under the inclusion of a psychological therapist might be helpful in adolescents with severe caries-related destruction of the permanent dentition. However, patients and/or parents often feel that this is not necessary.

Our study has methodological strengths and limitations. All study designs followed basic methodological recommendations for measuring dental health, which included calibration of the dentists, intraoral standard examination, and data exploration, as previously published [12,17,19,20]. In addition, due to the large number of dental recordings (N = 2875), each of the three study groups primarily followed a population-based recruitment strategy. Pertinently, the 10- and 15-year-olds and their families participated in a longitudinal cohort study from birth onwards and showed a long-term, above-average interest in this study project. The combined use of statistical measures, e.g., the SiC index with narrower thresholds of 20%, 10%, 5%, 2%, and 1%, the SaC index, and Lorenz curves, to illustrate the inequal distribution of caries in the adolescent population is a clear strength of the study and a methodological novelty.

As a potential limitation, by the 10-year follow-up, nearly half of the initially recruited subjects no longer participated [13]. This also applies to the cross-sectional dental examination in Bavaria, where the participation rate varied regionally. Therefore, the chosen cross-sectional data sets appear not to be representative. Furthermore, patients with a high or extremely high burden of caries may not be willing to participate in any observational, diagnostic, or epidemiological dental study, which may result in an underestimation of cases with severely damaged early permanent dentitions. The present statistical analyses included only dmft/DMFT data and excluded data about non-cavitated caries lesions, because these data were not available for primary dentition in 10-year-olds. Additionally, we used retrospective existing data with one dataset per age group. This data structure therefore did not allow for comparison between different time points or, due to different data sets, between age groups at the same time point. Individual factors that are known to influence the experience of caries in complex ways, such as socioeconomic, psychological, and behavioral factors of parents [30], have not been systematically investigated in the present studies. Because this study was more concerned with the methodological approach for investigation to access patients with a high caries risk, it should still be mentioned that the used data sets were not recently collected.

Nevertheless, the dmft/DMFT index and its methodology [16] have successfully become established as reliable diagnostic tools to determine the prevalence, experience, and polarization of caries [34]. To present the polarization of caries, previous studies focused on either a fixed age group, e.g., 12-year-olds [35] or 18-year-olds [36], to calculate DMFT and SiC values, or used Lorenz curves for 3-year-olds to show distributions while comparing different oral health systems [37]. Surprisingly, no epidemiological study known to us presents polarization of caries using the SiC and SaC indexes, in addition to the Lorenz curves, in the detail presented here, for these age groups.

## 5. Conclusions

On the basis of the present data, it can be concluded that the experience of caries was not equally distributed in the investigated adolescent populations. A minority of individuals accounted for the majority of the caries burden. Additionally, a very small group comprising mostly 15-year-olds was identified to have an extremely high caries burden (DMFT ≥ 8). With respect to the severe and long-term clinical consequences in permanent dentition for each of the affected individuals, further interdisciplinary research appears to be needed to clarify potential associations, and to develop successful caries prevention and management strategies for adolescents with an extreme caries risk.

## Figures and Tables

**Figure 1 ijerph-18-04878-f001:**
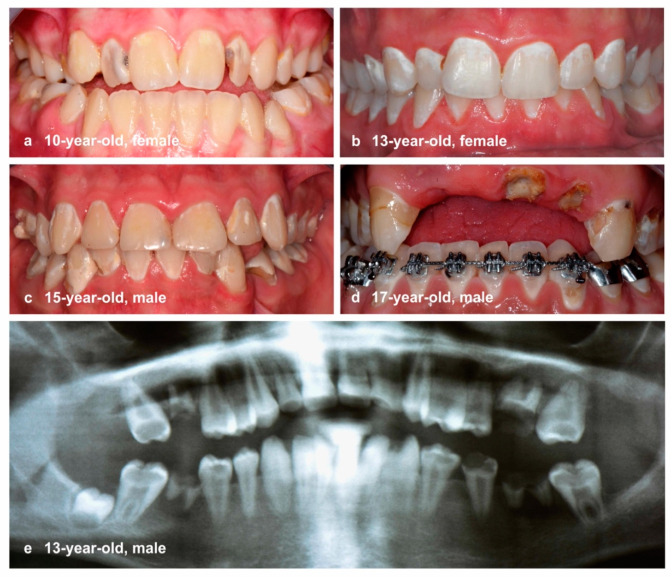
Clinical situation of severely caries-affected adolescents: multiple non-cavitated and cavitated caries lesions with typical additional individual risks: (**a**) mental health issues; (**b**) no medical history; (**c**) suspected uninvolved parenting (neglect); (**d**) suspected drug abuse and mental health issues; and (**e**) syndromic disease.

**Figure 2 ijerph-18-04878-f002:**
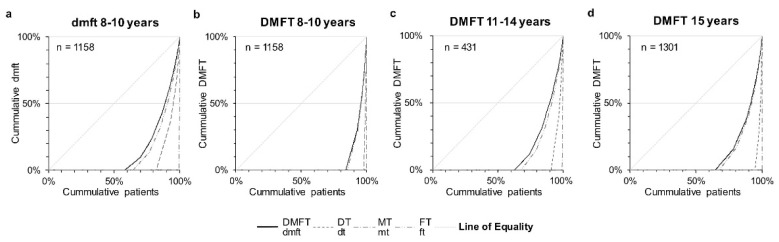
Lorenz curves of the dmft/DMFT values and their components of (**a**,**b**) 10-year-olds (N = 1158), (**c**) 12-year-olds (N = 431), and (**d**) 15-year-olds (N = 1301). The grey lines indicate the line of equality and 80% of the cumulative proportion of patients with caries burden.

**Table 1 ijerph-18-04878-t001:** Characteristics of investigated individuals in the three study populations.

	10-Year-Olds	12-Year-Olds	15-Year-Olds
Study	GINIplus + LISA	LAGZ	GINIplus + LISA
Study type	Cohort study	Survey study	Cohort study
Years	2005–2009	2016	2010–2014
Location	Munich	Bavaria	Munich
Number (N)	1158	416	1301
Mean (SD) age	10.2 (0.2)	12.1 (0.6)	15.2 (0.3)
N (%) male	523 (45.2%)	250 (60.0%)	652 (50.1%)
N (%) female	635 (54.8%)	166 (40.0%)	650 (50.0%)

**Table 2 ijerph-18-04878-t002:** Descriptive characterization of the caries experience and polarization according to the dmft/DMFT index, its components, the Significant Caries Index (SiC), and Specific Affected Caries Index (SaC) in relation to different threshold values.

Dentition	10-Year-Olds (N = 1158)				12-Year-Olds (N = 416)		15-Year-Olds (N = 1301)	
	Primary		Permanent		Permanent		Permanent	
**Caries prevalence**	N	%	N	%	N	%	N	%
Caries-free (dmft/DMFT = 0)	679	58.6	972	83.9	256	61.5	841	64.6
dmft/DMFT > 0	479	41.4	186	16.1	160	38.5	460	35.4
dmft/DMFT ≥ 4	155	13.4	23	2.0	39	9.4	108	8.3
dmft/DMFT ≥ 8	12	1.0	0	0	1	0.2	8	0.6
Caries-affected, without restorations (dmft/DMFT > 0 AND ft/FT = 0)	75	6.5	13	1.1	21	5.0	40	3.1
**Caries experience**	**Mean (SD)**		**Mean (SD)**		**Mean (SD)**		**Mean (SD)**	
dmft/DMFT	1.2 (1.9)		0.3 (0.8)		1.0 (1.6)		1.0 (1.6)	
dt/DT	0.3 (0.8)		0.0 (0.2)		0.2 (0.6)		0.1 (0.6)	
mt/MT	0.0 (0.1)		0.0 (0.0)		0.0 (0.2)		0.0 (0.1)	
ft/FT	0.9 (1.6)		0.3 (0.7)		0.8 (1.5)		0.9 (1.6)	
**Caries polarization**	**N**	**Mean (SD)**	**N**	**Mean (SD)**	**N**	**Mean (SD)**	**N**	**Mean (SD)**
Caries experience in subjects with dmft/DMFT > 0 (SaC)	479	2.9 (1.9)	186	1.8 (1.1)	160	2.6 (1.7)	460	2.5 (1.8)
dmft/DMFT of upper 33% (SiC)	382	3.4 (1.8)	386	0.9 (1.2)	137	2.8 (1.7)	434	2.5 (1.8)
dmft/DMFT of upper 20%	232	4.4 (1.5)	232	1.4 (1.2)	83	3.7 (1.7)	260	3.5 (1.8)
dmft/DMFT of upper 10%	116	5.5 (1.4)	116	2.3 (1.1)	42	4.7 (1.8)	130	4.8 (1.7)
dmft/DMFT of upper 5%	58	6.6 (1.2)	58	3.1 (0.9)	21	5.7 (2.3)	65	6.0 (1.7)
dmft/DMFT of upper 2%	23	7.7 (1.1)	23	4.1 (0.3)	8	7.1 (3.0)	26	7.5 (1.8)
dmft/DMFT of upper 1%	12	8.5 (1.0)	12	4.2 (0.4)	4	8.5 (3.8)	13	8.5 (2.0)

**Table 3 ijerph-18-04878-t003:** Detailed distribution for the components of the dmft/DMFT index in the proportion of most affected patients regarding dmft/DMFT values (upper 33%–1%).

	Mean Values for the Components of the dmft/DMFT Index for the SiC and Other Thresholds											
	Upper 33% (SiC)		Upper 20%		Upper 10%		Upper 5%		Upper 2%		Upper 1%	
N 10-year-olds	382		232		116		58		23		12	
Mean dt (SD)/%	0.9 (1.2)	26.6	1.1 (1.4)	24.1	1.5 (1.6)	27.1	1.6 (1.9)	24.9	1.5 (2.0)	19.7	1.5 (2.2)	17.6
Mean mt (SD)/%	0.0 (0.1)	0.4	0.0 (0.2)	0.5	0.0 (0.2)	0.6	0.0 (0.2)	0.5	0.0 (0.0)	0.0	0.0 (0.0)	0.0
Mean ft (SD)/%	2.5 (1.9)	73.0	3.3 (1.9)	75.4	4.0 (2.2)	72.2	4.9 (2.4)	74.6	6.2 (2.3)	80.3	7.0 (2.2)	82.4
Mean DT (SD)/%	0.1 (0.3)	8.0	0.1 (0.3)	8.0	0.2 (0.4)	7.1	0.2 (0.4)	5.6	0.2 (0.4)	5.3	0.2 (0.4)	4.2
Mean MT (SD)/%	0.0 (0.1)	0.3	0.0 (0.1)	0.3	0.0 (0.1)	0.3	0.0 (0.0)	0.0	0.0 (0.0)	0.0	0.0 (0.0)	0.0
Mean FT (SD)/%	0.8 (1.1)	91.7	1.3 (1.2)	91.7	2.1 (1.1)	92.5	2.9 (0.9)	94.4	3.9 (0.3)	94.7	4.0 (0.0)	96.0
N 12-year-olds	137		83		42		21		8		4	
Mean DT (SD)/%	0.4 (0.9)	14.9	0.5 (1.0)	13.4	0.6 (1.3)	13.7	0.7 (1.4)	12.8	1.0 (1.8)	14.0	0.8 (1.3)	8.8
Mean MT (SD)/%	0.1 (0.4)	1.8	0.1 (0.5)	2.0	0.1 (0.6)	2.0	0.2 (0.9)	3.4	0.0 (0.0)	0.0	0.0 (0.0)	0.0
Mean FT (SD)/%	2.2 (1.5)	79.6	2.3 (1.9)	84.6	4.0 (2.3)	84.3	4.7 (2.9)	98.0	6.1 (3.8)	86.0	7.8 (4.4)	91.2
N 15-year-olds	434		260		130		65		26		13	
Mean DT (SD)/%	0.3 (0.9)	11.1	0.4 (1.1)	11.1	0.5 (1.4)	11.0	0.6 (1.7)	9.3	1.0 (2.4)	12.8	1.2 (2.8)	13.5
Mean MT (SD)/%	0.0 (0.1)	0.3	0.0 (0.1)	0.3	0.0 (0.2)	0.5	0.0 (0.1)	0.3	0.0 (0.0)	0.0	0.0 (0.0)	0.0
Mean FT (SD)/%	2.3 (1.8)	88.6	3.1 (1.9)	88.6	4.3 (2.0)	88.5	5.4 (2.5)	90.5	6.5 (2.5)	87.2	7.4 (2.7)	86.5

dt/DT: decayed primary/permanent teeth; mt/MT: missing primary/permanent teeth; ft/FT: filled primary/permanent teeth.

## Data Availability

The datasets used and/or analyzed during the study are available from the corresponding author upon request.

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
