# Peer review of "Distribution and Polarization of Caries in Adolescent Populations"

_ijerph, 2021, doi:10.3390/ijerph18094878_

Round 1
Reviewer 1 Report
This study is a valuable survey on the prevalence of dental caries in Germany using a large data set. However, the following points need to be considered.
- P2L57-59: Please add the reference.
- P3L74: Please add the explain for the " inclusion and exclusion criteria " in the text.
- P5L171-183: I think this content falls under the discussion.
- Since the purpose of this study is to understand the current dental caries prevalence situation in Germany, which differs from previous trends, the discussion should take into account the situation in Germany to date.
- Many studies have shown that factors such as socioeconomic status are involved in the background of extreme disparities in dental caries prevalence. Therefore, background other than individual factors such as socioeconomic status should be considered using previous studies. In addition, the possibility of the existence of such confounding factors should be described as a limitation of the study.
Author Response
Revision 1 – open review
Point by point answers
Reviewer #1
This study is a valuable survey on the prevalence of dental caries in Germany using a large data set. However, the following points need to be considered.
Thank you for your discussion points and suggestions. Building on your remarks, we have extensively revised the manuscript to meet the criteria.
- P2L57-59: Please add the reference.
Answer: Thank you, we added the reference accordingly. In general, the prevalence of caries in adolescents has been declining in recent decades - as has been shown in particular in epidemiological studies on German oral health and it also reflects our practical experience.
Revision in manuscript: Please see line 47. - P3L74: Please add the explain for the "inclusion and exclusion criteria " in the text.
Answer: Thank you for your suggestion. The description of the inclusion and exclusion criteria are taken directly from the respective publications, translated and compiled in the following table for a better overview.
The basic inclusion and exclusion criteria are added have already been extensively described and discussed: GINIplus and LISA already in the paper Heinrich et al. (2012). LAGZ is described in detail (Uebereck C, 2017). In the manuscript, references have been focused on the main articles on the inclusion and exclusion criteria, for reasons of redundancy, a renewed detailed discussion has been omitted.
Revision in manuscript: Please see line 105-111. - P5L171-183: I think this content falls under the discussion.
Answer: Thank you for noticing. We placed it more appropriately in the discussion section.
Revision in manuscript: Please see line 419-433. - Since the purpose of this study is to understand the current dental caries prevalence situation in Germany, which differs from previous trends, the discussion should take into account the situation in Germany to date.
Answer: Thank you for your consideration. In particular, we have used this article to examine the striking disparity in caries distribution in adolescents - in particular, clinically, we see clustered cases with extraordinary caries experience that seem to be elusive in typical epidemiologic studies. We have added relevant posts.
Revision in manuscript: Please see line 21-23, 77-78. - Many studies have shown that factors such as socioeconomic status are involved in the background of extreme disparities in dental caries prevalence. Therefore, background other than individual factors such as socioeconomic status should be considered using previous studies. In addition, the possibility of the existence of such confounding factors should be described as a limitation of the study.
Answer: Thank you for your consideration. Current epidemiological studies do not adequately represent these single individuals with extreme caries experience, so our goal was to be able to map this group. However, our study data do not reflect factors such as socioeconomic status that certainly contribute to caries experience. We have elaborated on these factors in the discussion and limitations.
Revision in manuscript: Please see line 465-468.
Heinrich, J., Brüske, I., Schnappinger, M., Standl, M., Flexeder, C., Thiering, E., Koletzko, S. (2012). [Two German Birth Cohorts: GINIplus and LISAplus]. Bundesgesundheitsblatt, Gesundheitsforschung, Gesundheitsschutz, 55(6-7), 864-874. doi:10.1007/s00103-012-1485-4
Uebereck C, K. h. J., Michel R, Taschner M, Frankenberger R, Krämer N. (2017). Zahngesundheit bayerischer Schulkinder 2015/16. Oralprophylaxe Kinderzahnheilkd, 39, 161- 171.

Reviewer 2 Report
This research is under the scope of this journal; the topic is relevant for readers, and this research deals with potentially significant knowledge to the field.
Introduction: What is the importance of this study? You do not think this study is included to the others already done? Which results are comparable?
"Nevertheless, while the population-wide caries burden has fallen, it is also a well-known epidemiological trend that caries is not equally distributed over the population, typically affect ing groups or individuals at elevated risks" Special needs patients should be considered. Please read : doi: 10.3390/jcm10020182.
The aim is unclear
Lack of significance (p <???) in the explanations, in the results section
There are many mistakes in the references section and in the text
The discussion is also misleading. What is the novelty of this paper???
Limitations?
Conclusions were not totally supported by the data showed.
Figure legends: Bad descriptions
Author Response
Revision 1 – open review
Point by point answers
Reviewer #2
This research is under the scope of this journal; the topic is relevant for readers, and this research deals with potentially significant knowledge to the field.
Thank you for your review and your suggestions. We hope to answer your questions sufficiently and that we’ve been able to implement your comments in a meaningful way for better understanding of our manuscript.
- Introduction: What is the importance of this study? You do not think this study is included to the others already done? Which results are comparable?
Answer: Thank you for your question. The focus of the study is explicitly on the extent of inequality and the "frontrunners" in caries experience. Fortunately, the epidemiological studies known to us show the well-known trend of lower caries experience in recent decades due to sufficiency of prevention strategies. Nevertheless, in our clinical practice we repeatedly see patients with massive caries experience in whom all prevention strategies fail. We have elaborated on this point at various points in the manuscript.
Revision in manuscript: Please see lines 33-35, 75-81. - "Nevertheless, while the population-wide caries burden has fallen, it is also a well-known epidemiological trend that caries is not equally distributed over the population, typically affecting groups or individuals at elevated risks" Special needs patients should be considered. Please read: doi: 10.3390/jcm10020182.
Answer: Thank you for this important note. Indeed, massive findings are frequently found in patients with special needs. Often such patients are additionally excluded from epidemiological studies - as in the ones presented here - so that their urgent need is little investigated and perceived despite ethical obligation.
Revision in manuscript: Please see lines 47-64. - The aim is unclear
Answer: Thank you. We specified the aim in more detail.
Revision in manuscript: Please see line 21-23. - Lack of significance (p <???) in the explanations, in the results section
Answer: Thank you for your concern. Since we retrospectively examined already collected data sets and had only one data set per for 10-, 12-, 15-year-olds each, we unfortunately could not draw a comparison and therefore could not calculate significant values (p-values). It could be an interesting point to calculate older data sets with our method. We added this limitation to our discussion section.
Revision in manuscript: Please see lines 456-457. - There are many mistakes in the references section and in the text
Answer: Thank you for your note. We double-checked and corrected the citation section.
Revision in manuscript: Please see references. - The discussion is also misleading. What is the novelty of this paper???
Answer: Thank you for your comment. The use of Sic/SaC Indices and Lorenz curves to display the polarization of caries were a new approach to represent patients with exceptionally high caries experience also in epidemiological studies. In addition to the SiC value we used thresholds such as 20%, 10%, 5%, 2% and 1% of the dmft/DMFT to point out precisely the unequal caries distribution.
Revision in manuscript: Please see line 523-526. - Limitations?
Answer: Thank you for your concern – we restructured and discussed in more detail the potential limitations.
Revision in manuscript: Please see lines 527-542. - Conclusions were not totally supported by the data showed.
Answer: Thank you for your note – we specified the conclusions.
Revision in manuscript: Please see lines 553-560. - Figure legends: Bad descriptions
Answer: Thank you for your advice – we improved the quality of the individual photos compiled in Figure 1 and additionally optimized the legend for a clearer display.
Revision in manuscript: Please see Figure 1.

Reviewer 3 Report
Thank you for choosing me as a reviewer of this manuscript. In my opinion, the topic of this manuscript is important and interesting. The problem of the intensity of caries is always actual and the introduction section is reliable and comprehensive with an actual review of the literature. I recommend the manuscript for publication, but after some major revisions:
1.Discussion section must be improve-in my opinion authors should consider the issue wider. Line 194-214- This part doesn’t belong to the discussion-in my opinion it is a description of the results
2.I’m a little confused: I'm not sure if I understand correctly, but the study is based on data collected previously? Could you explain it to me? Because as far I noticed research was conducted in 2005-2009; 2010-2014 and 2016? So that’s why I’m a little concerned about the actuality of this research?
Author Response
Revision 1 – open review
Point by point answers
Reviewer #3:
Thank you for choosing me as a reviewer of this manuscript. In my opinion, the topic of this manuscript is important and interesting. The problem of the intensity of caries is always actual and the introduction section is reliable and comprehensive with an actual review of the literature. I recommend the manuscript for publication, but after some major revisions:
Thank you for your valuable comments regarding our article. We hope to answer your questions sufficiently and that we’ve been able to incorporate your comments in a meaningful way for better understanding of our manuscript.
- Discussion section must be improved-in my opinion authors should consider the issue wider. Line 194-214- This part doesn’t belong to the discussion-in my opinion it is a description of the results
Answer: Thank you for your note. We revised our discussion, accordingly.
Revision in manuscript: Please see discussion section. - I’m a little confused: I'm not sure if I understand correctly, but the study is based on data collected previously? Could you explain it to me? Because as far I noticed research was conducted in 2005-2009; 2010-2014 and 2016? So that’s why I’m a little concerned about the actuality of this research?
Answer: Correct, instead of conducting a separate investigation on the issue, we used existing data sets to answer the given research question. Additionally, we added the actuality of the used data set from previously conducted trials as potential limitation.
Revision in manuscript: Please see lines 535-543.

Reviewer 4 Report
“Caries distribution and polarization in adolescent populations”
In the manuscript, authors analyze caries prevalence, experience and polarization prevalence in adolescent populations and determine the proportion with severe caries decay. The paper is well directed and written and follows a well stablished methodology. However, the authors may attend some suggestion after to be considered for publication in IJERPH.
Abstract.
- In Line 21-22 “The aim of this study was to determine the proportion of 12- and 15-year-olds with severe caries decay to analyse caries prevalence, experience and polarization. So what is the reason for studying the 10-year-old population? Please explain the reason.
- Also, the abstract only presents the summary of the results of the statistical study (mean dmf / DMF values (standard deviation), Significant Caries Index (SiC) values and Specific Affected Caries Index (SaC) values and Lorenz curves) and does not presents an analysis of relevance or a conclusion of the analysis of these results.
Introduction.
- According to the authors, Lines 43-45 ”prevalence and experience of documented caries declined steadily over time, reaching an all-time low, especially in younger generations who likely benefited substantially from individual and group preventive programs…” and Lines 57-61 “Knowing full-well that consultations regarding adolescents with severely affected permanent dentition might be exceptional in German dental practices currently, it seems to be of relevance to determine the proportion and extent of the hypothesized unequal distribution from an epidemiological point of view.” The aim of the present study is unclear. Clarify.
Discussion
- Line 256-258…..“Surprisingly, no epidemiological study known to us presents caries polarization using the SiC and SaC as well as Lorenz curves in such detail in these age groups”…..It must remarked in the introductory section as a contribution on the field.
Author Response
Revision 1 – open review
Point by point answers
Reviewer #4:
“Caries distribution and polarization in adolescent populations”
In the manuscript, authors analyze caries prevalence, experience and polarization prevalence in adolescent populations and determine the proportion with severe caries decay. The paper is well directed and written and follows a well stablished methodology. However, the authors may attend some suggestion after to be considered for publication in IJERPH.
Thank you for your thoughtful comments. We hope that we’ve been able to answer all questions and properly incorporate the suggested changes in the manuscript.
- In Line 21-22 “The aim of this study was to determine the proportion of 12- and 15-year-olds with severe caries decay to analyse caries prevalence, experience and polarization. So what is the reason for studying the 10-year-old population? Please explain the reason.
Answer: Thank you for your comment. Due to the limited word count in the abstract section (N=200), we changed the wording. Moreover, we analyzed the group of 10-year-olds, because the dataset was available.
Revision in manuscript: Please see lines 21-23. - Also, the abstract only presents the summary of the results of the statistical study (mean dmf / DMF values (standard deviation), Significant Caries Index (SiC) values and Specific Affected Caries Index (SaC) values and Lorenz curves) and does not presents an analysis of relevance or a conclusion of the analysis of these results.
Answer: Thank you for your comment. Considering the limited word count in the abstract, we tried to provide as much information as possible. We revised the abstract with the goal of strengthening the statement for clinical relevance.
Revision in manuscript: Please see lines 21-33. - According to the authors, Lines 43-45 ”prevalence and experience of documented caries declined steadily over time, reaching an all-time low, especially in younger generations who likely benefited substantially from individual and group preventive programs…” and Lines 57-61 “Knowing full-well that consultations regarding adolescents with severely affected permanent dentition might be exceptional in German dental practices currently, it seems to be of relevance to determine the proportion and extent of the hypothesized unequal distribution from an epidemiological point of view.” The aim of the present study is unclear. Clarify.
Answer: Thank you for your comment. We tried to clarify the aim and revised the introduction section for better understanding.
Revision in manuscript: Please see lines 21-23.
- Discussion
Line 256-258…..“Surprisingly, no epidemiological study known to us presents caries polarization using the SiC and SaC as well as Lorenz curves in such detail in these age groups”…..It must remarked in the introductory section as a contribution on the field.
Answer: Thank you for your remark. We added this to our introduction section.
Revision in manuscript: Please see lines 76-81.

Reviewer 5 Report
The original data sets are a valuable tool for the presented review. Please discuss why the MO Lorenz` methods measuring the concentration of wealth are applied to the dental caries prevalence? Please also improve the Fig.1: b and d are too dark, replace e with a much better radiographic example. Discussion needs improvement: Known aetiology is a very complex field of extrinsic and intrinsic factors of caries susceptibility, presented exclusively from the narrow epidemiological point of view. Therefore, avoid pure speculations (216 - 233) and oversimplification of aetiology.
Please follow the common instructions to authors and correct the References in one style.
Author Response
Revision 1 – open review
Point by point answers
Reviewer #5:
Thank you, for your considerate review. We hope that we’ve been able to answer open questions and properly integrate the recommended changes in the article.
- The original data sets are a valuable tool for the presented review. Please discuss why the MO Lorenz` methods measuring the concentration of wealth are applied to the dental caries prevalence?
Answer: Thank you for your question. Since the traditional use of the DMF index is not adequate to represent inequalities and Lorenz curve is not a standard for visualizing the unequal distribution in epidemiological studies in dentistry, we contribute hereby to establish it. Therefore, we took the advantage of illustrating inequality using the Lorenz curve with cumulative DMF measures. We added this to our discussion section.
Revision in manuscript: Please see lines 76-81, 540-543. - Please also improve the Fig.1: b and d are too dark, replace e with a much better radiographic example.
Answer: Thank you for this advice – we improved the quality of the individual photos compiled in Figure 1 and additionally optimized the legend for a clearer display.
Revision in manuscript: Please see Figure 1. - Discussion needs improvement: Known aetiology is a very complex field of extrinsic and intrinsic factors of caries susceptibility, presented exclusively from the narrow epidemiological point of view. Therefore, avoid pure speculations (216 - 233) and oversimplification of aetiology.
Answer: Thank you for your remark. We improved the mentioned discussion section.
Revision in manuscript: Please see lines 461-465.

Round 2
Reviewer 3 Report
Dear authors,
Thank you for your efforts to improve the article. I have no more comments